# MEXPLORER 1.0.0 – A mechanism explorer for analysis and visualization of chemical reaction pathways based on graph theory

**Rolf Sander**

Air Chemistry Department, Max-Planck Institute of Chemistry, P.O. Box 3060, 55020 Mainz, Germany

**Abstract.** The open-source software MEXPLORER 1.0.0 is presented here. The program can be used to analyze, reduce, and visualize complex chemical reaction mechanisms. The mathematics behind the tool is based on graph theory: chemical species are represented as vertices, and each reaction is described as a set of edges. MEXPLORER is a community tool published under the GNU General Public License.

## 1 Introduction

When research in atmospheric chemistry started about a century ago, only a couple of reactions were known. For example, Chapman (1930) identified the most important reactions explaining the stratospheric ozone layer:

$$O_3 + O \quad \rightarrow \quad 2\,O_2 \tag{R1}$$
$$O_2 + O \quad \rightarrow \quad O_3 \tag{R2}$$
$$O_2 \quad \overset{h\nu}{\rightarrow} \quad 2\,O \tag{R3}$$
$$O_3 \quad \overset{h\nu}{\rightarrow} \quad O_2 + O \tag{R4}$$

Since then, numerous additional chemicals have been discovered in the atmosphere, and our knowledge about their reactions has grown immensely. Today, many models are available to describe the atmospheric degradation of organics in the gas and the aqueous phase, e.g., the Mainz Organic Mechanism (MOM, Sander et al., 2019) and the Jülich Aqueous-phase Mechanism of Organic Chemistry (JAMOC, Rosanka et al., 2021). The most comprehensive set of reactions for tropospheric organic compounds is the Master Chemical Mechanism (MCM), which contains more than 15 000 reactions[1].

Several methods have been developed to reduce the complexity of such large mechanisms. The approach to combine chemical species manually into families has been used for a long time (Crutzen and Schmailzl, 1983). Newer approaches include the Common Representative Intermediates (CRI) by Watson et al. (2008) and the skeletal mechanism reduction (Tomlin and Turányi, 2013). Such automated methods produce smaller mechanisms which can increase the speed of numerical simulations. However, when used as a black box, they do not contribute to our understanding of the system. Here, the MEXPLORER (mechanism explorer) software is presented, which contains several tools to identify, analyze and visualize important parts of complex reaction mechanisms. Its main features are described in the following sections. Additional information is provided in the user manual, which contains detailed instructions how to install and use the code.

Although this work focuses on atmospheric chemistry, the software can be used in other fields as well, e.g., combustion chemistry (Westbrook et al., 2009) or the automated generation of complex mechanisms (e.g., Martínez-Núñez et al., 2021; Garay-Ruiz et al., 2022; Sumiya et al., 2022; Maeda et al., 2023).

From a mathematical point of view, a set of chemical reactions can be seen as a directed graph (digraph). A detailed description how graph theory can be used to analyze kinetic reaction mechanisms is presented in the book by Turányi and Tomlin (2014).

## 2 Description

There are several ways to define a digraph of a chemical mechanism, for example with a bipartite "species-reaction graph" where both, species and reactions, are represented by vertices (e.g., Silva et al., 2020). An alternative is the "kinetics graph" where each reaction is represented as a set of edges pointing from all reagents to all products. MEXPLORER uses the latter approach. Especially for the visualization of complex mechanisms (see below), this produces plots with much less visual clutter. As long as the reaction

---

[1]https://mcm.york.ac.uk/MCM

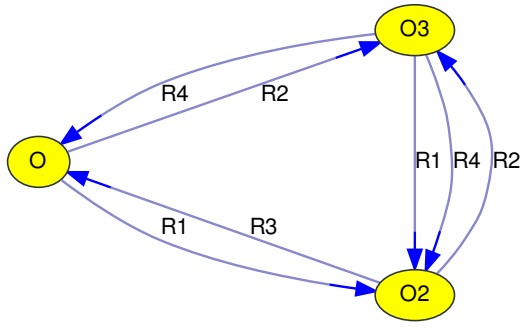

**Figure 1.** Graph of the Chapman mechanism, created with MEX-PLORER.

corresponding to each edge is stored as an edge property, both graph types contain the same information. To convert a kinetics graph into a species-reaction graph, it is only necessary to insert nodes representing reactions into the edges
5 between reagents and products.

Taking the above-mentioned Chapman mechanism as an example, a graph with 3 vertices ($V_1 \ldots V_3$) and 7 edges ($E_1 \ldots E_7$) can be constructed:

| | | | | |
|---|---|---|---|---|
| $V_1$: | O | | | |
| $V_2$: | $O_2$ | | | |
| $V_3$: | $O_3$ | | | |
| $E_1$: | O | $\rightarrow$ | $O_2$ | (R1) |
| $E_2$: | O | $\rightarrow$ | $O_3$ | (R2) |
| $E_3$: | $O_2$ | $\rightarrow$ | O | (R3) |
| $E_4$: | $O_2$ | $\rightarrow$ | $O_3$ | (R2) |
| $E_5$: | $O_3$ | $\rightarrow$ | O | (R4) |
| $E_6$: | $O_3$ | $\rightarrow$ | $O_2$ | (R1) |
| $E_7$: | $O_3$ | $\rightarrow$ | $O_2$ | (R4) |

10    As input, MEXPLORER needs a text file defining the chemical species and the mechanism in KPP format (Sandu and Sander, 2006). For the Chapman mechanism, the species and their elemental composition are defined as:

```
   O  =   O ;
15 O2 = 2O ;
   O3 = 3O ;
```

The reactions are defined as:

```
   <R1> O3 + O  = 2 O2   : k1;
   <R2> O2 + O  = O3     : k2;
20 <R3> O2 + hv = 2 O     : k3;
   <R4> O3 + hv = O + O2  : k4;
```

Many atmospheric-chemistry models are already using KPP, thus, importing their mechanisms into MEXPLORER is straightforward. Once the mechanism has been stored in

the graph-tool-specific xml format, several tools and algo-     25
rithms from graph theory can be applied in order to analyze, reduce and visualize (Fig. 1) the chemical mechanism, as discussed in the following sections.

The installation requires version 3.6 of Python[2] and graph-tool[3]. Drawing the graphs is based on Graphviz[4]. Execution     30
of the code is controlled via configuration ("config") files, using the Python ConfigParser.

## 2.1    Visualization

MEXPLORER can create a plot of the whole mechanism or a selected subset and save it as a vector graphic in pdf format.     35
According to "a picture is worth a thousand words", such plots can provide an informative overview for presentations and publications. The user can adjust several features of the plot.

### 2.1.1    Vertices (species)     40

Chemical species are represented as vertices of the graph. To increase the amount of information contained in the plot, MEXPLORER offers several options to tune the background color and labels of the vertices:

– For plots showing the pathways from a source species     45
  to a target species, this can be emphasized with a red background color for the source and a green background color for the target. Intermediates in the path are shown with a yellow background color (see Fig. 2).

– To illustrate the degradation of large organic molecules     50
  in a chain of reactions, the number of carbon atoms can be used to determine the background color, as in Fig. 3.

– In a multiphase chemical mechanism, different colors can be used for gas- and aqueous-phase species, respectively. This coloring option is used in Fig. 4, where the     55
  aqueous-phase species contain the suffix "_a01".

– Another area of application is the skeletal reduction of a complex mechanism to a simpler mechanism. A detailed description of the methodology has been presented by Niemeyer and Sung (2011). Briefly, all reac-     60
  tion sequences from a source to a target species in the mechanism are analyzed. An overall interaction coefficient (OIC) is calculated for every species in the mechanism, where a high OIC value indicates an important species. Species with an OIC below a certain threshold     65
  are removed from the mechanism. MEXPLORER can read such OIC values from a file and use them to define the vertex background color. In addition, the OIC values can be shown inside the vertices below the names of

---

[2]https://www.python.org/

[3]https://graph-tool.skewed.de

[4]https://graphviz.org

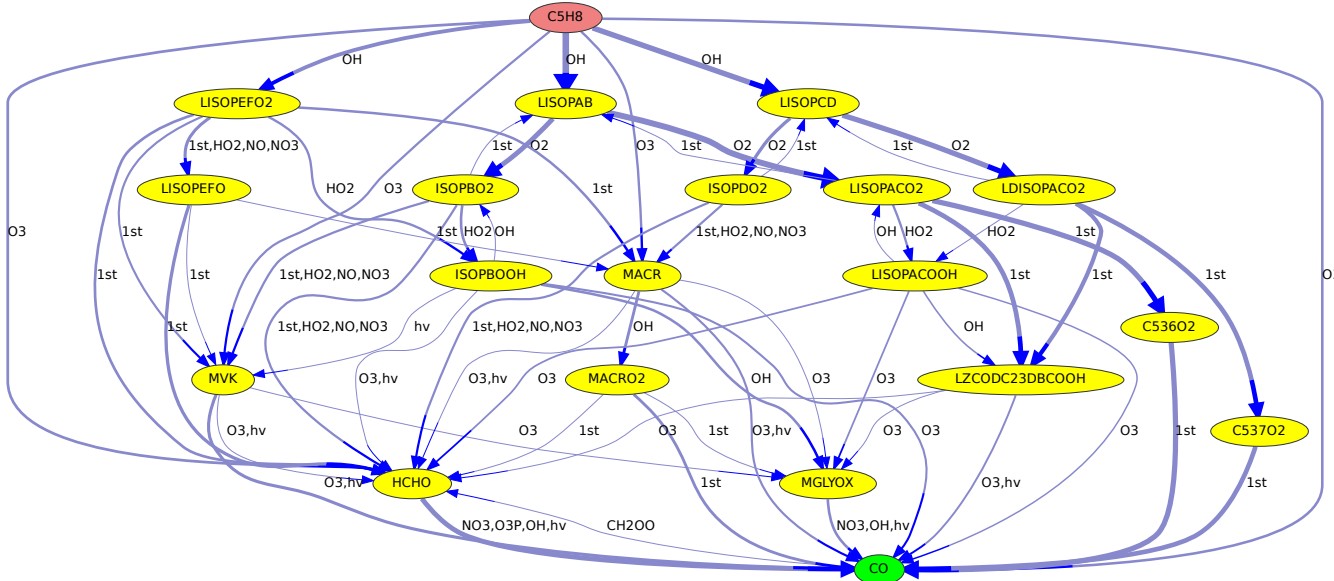

**Figure 2.** Reactions from $C_5H_8$ (isoprene) to CO in the MOM mechanism. Line thickness corresponds to reaction rates.

the species. This produces a clear representation of the reduced mechanism (see for example Fig. 4 in Sander et al. (2019)).

All of the options listed above can be controlled via Python config files. For alternative representations of the vertex properties, the code provides an entry point for additional, user-defined Python functions.

### 2.1.2 Edges (reactions)

A chemical reaction is represented as a set of directed edges in the graph. To increase the amount of information contained in the plot, MEXPLORER offers several options to tune the thickness, color and labels of the edges:

- When reaction rates are available, they can be used to define the line thickness of the edge arrows. This is especially useful to highlight important reactions found in maximum-flow calculations, as in Figs. 2 and 4.

- Edge labels are used to identify reactions. In Fig. 1, the reaction numbers (R1,...,R4) are shown. Although this is a unique and consise way to describe edges, it requires the look-up of the reactions that correspond to these reaction numbers. An alternative is to label edges with the other reactants of the reaction (as in Figs. 2, 3, 4, and 5). For photochemical reactions, the label "$h\nu$" is used, and first-order reactions are labeled with "1st". Unfortunately, the graphs quickly become unreadable with increasing complexity of the chemical mechanism. To alleviate this problem, MEXPLORER can merge parallel edges and their labels. For example,

the Chapman mechanism in Fig. 1 contains two reactions that convert $O_3$ to $O_2$: R1 and R4. After merging the corresponding edges, a new label with a comma-separated list is created: "R1,R4". When the edge labels become too long, they are abbreviated to "n rxns", indicating that there are $n$ reactions contributing to this edge. Examples of labels generated this way can be seen in Figs. 2, 3, 4 and 5.

### 2.1.3 Filters (reduced mechanism)

Plotting the complete graph of a complex mechanism will be unreasonable in many cases, making it necessary to extract a subset in order to obtain a clearly arranged plot. Available filters are:

- It is possible to select only species containing a specified element. For example, organic chemistry is selected with "element=C".

- From a multiphase mechanism, a specific phase can be selected, e.g., "phase=gas" or "phase=aqueous".

- Species that should not appear in a plot can be put into a blacklist, e.g., "blacklist=CO". Likewise, species that should always appear in the plot, even though they have already been removed by another filter, can be put into a whitelist, e.g., "whitelist=CO2".

- It is possible to remove slow reactions, i.e., all edges below a specified reaction rate, e.g., "minrxnrate=1E-19".

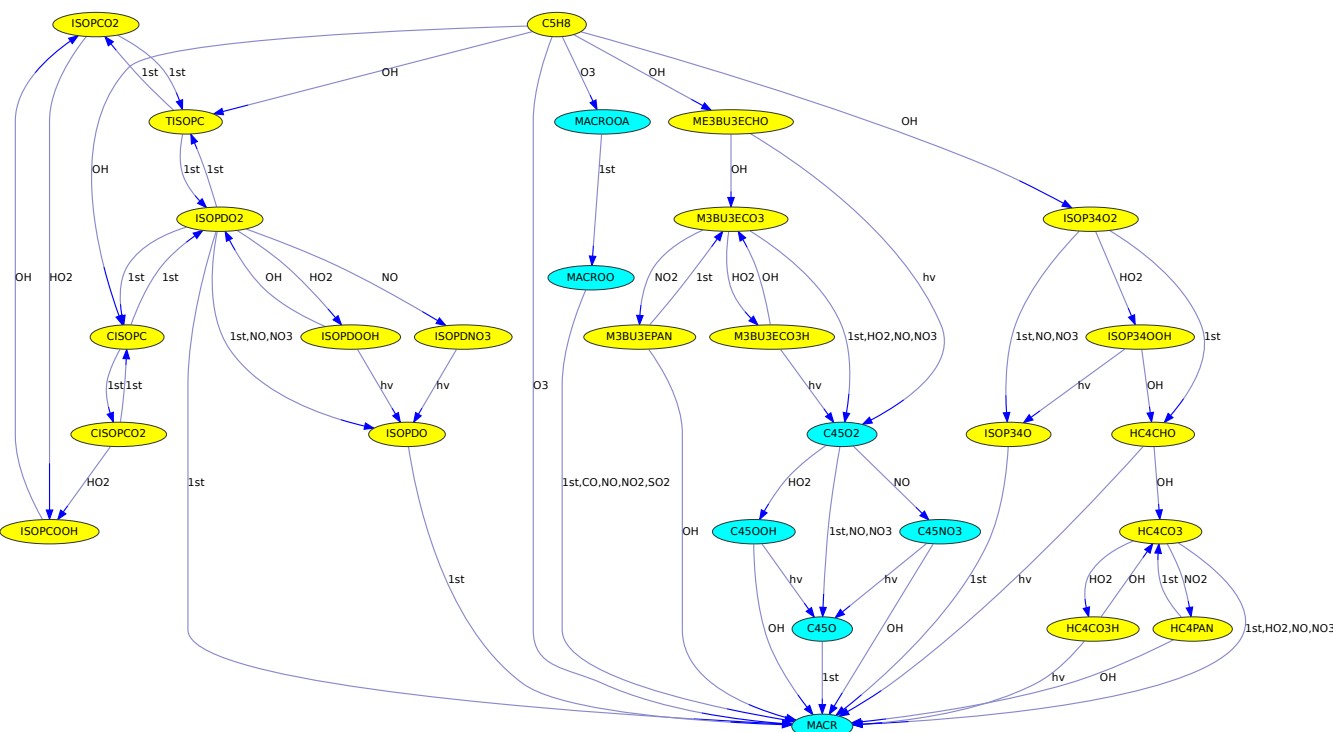

**Figure 3.** Reactions from $C_5H_8$ (isoprene) to MACR (methacrolein) in the MCM. Different colors are used for species with 5 and 4 carbon atoms.

– When studying the degradation of organic molecules, it is often helpful to remove all edges where the number of C atoms increases, using "`l_filter_out_C_increase=True`". See Sect. 2.2 for details.

– The max-flow algorithm (Sect. 2.3.1) filters out all reactions that do not contribute to the maximum flow from a source species to a target.

– Filters for special cases can be created by adding individual functions to the Python code. For example, MEXPLORER includes a function for the JAMOC mechanism (Rosanka et al., 2021), which excludes all gas-phase species, all inorganic species, and all species that don't have any chemical sources or sinks in the aqueous phase (only transfer from the gas phase). The option is activated with "`vfilter=jamoc`". This filter can serve as a template when a new, user-defined function needs to be written for a specific task.

### 2.1.4   Interactive visualization

There is also an option to explore a mechanism interactively in a GTK+ window. It is possible to pan, zoom or rotate the graph. A group of vertices can be selected to drag or rotate them together. For reordering the vertices, the dynamic spring-block layout can be activated. Sources and sinks of selected species can be shown. Additional Python functions can be written to activate more keystrokes in the interactive window.

### 2.2   Mechanism development

MEXPLORER can be used as a tool during mechanism development. Several sanity checks can be performed quickly while a new mechanism is under construction.

– MEXPLORER can show precursors and successors of selected species. As an example, the graph in Fig. 5 was created with the command "`neighbors=VINOH,1,1`". It shows all species in MOM that are directly connected to ethenol (vinyl alcohol, VINOH). Such a visualization can help to find missing sources or sinks while a mechanism is under construction. Using larger numbers, e.g., "`neighbors=VINOH,2,3`" creates a more complex graph with all species that can produce VINOH within 2 steps and all species that are produced from VINOH within 3 steps.

– MEXPLORER can find all reaction pathways from a source to a target species. Fig. 3 was produced with "`vfilter=src_tgt`", "`src=C5H8`", and "`tgt=MACR`". It shows the reaction chain transforming isoprene into methacrolein.

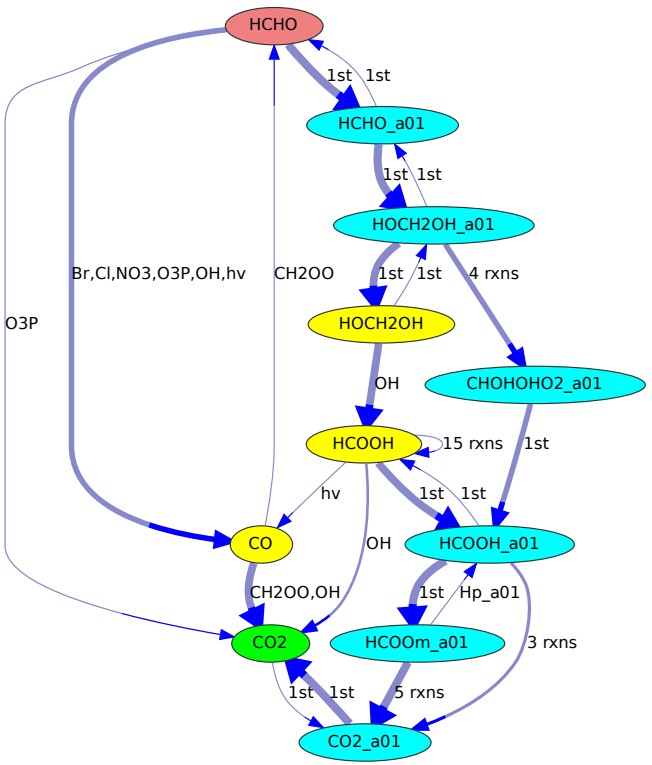

**Figure 4.** Reaction pathways from HCHO (red) to $CO_2$ (green) in JAMOC. The oxidation takes place in both the gas phase (yellow species) and also in the aqueous phase (blue species with the suffix "_a01"). Line thickness corresponds to reaction rates.

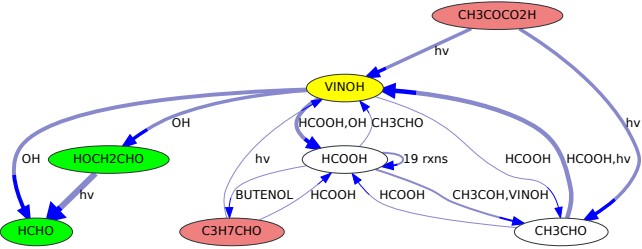

**Figure 5.** Sources and sinks of ethenol (VINOH, yellow) in MOM. Species that can produce VINOH are shown in red, and species that are produced from VINOH are shown in green. Species for which both applies are shown on a white background. Line thickness corresponds to reaction rates.

Often, a mechanism is created to study the degradation of large organic molecules with a monotonic decrease of the number of carbon atoms with each reaction step ($\Delta C \leq 0$). To remove all edges in which the number of carbon atoms increases ($\Delta C > 0$), "l_filter_out_C_increase=True" can be defined in the config file. Without such filtering, edges like $CO \rightarrow RO\cdot$ from the reactions of peroxy radicals with carbon monoxide ($RO_2\cdot + CO \rightarrow RO\cdot + CO_2$) would remain in the graph, adding even more reactions and unwanted complexity to the plot.

– While creating a graph from the KPP input files, the mechanism can be checked for unintentional mass balance violations. Specific elements are selected with e.g., "mass_balance=C,N,Cl". Of course, this feature requires that the elemental composition of the chemicals has been defined in the input file.

– For most species in a mechanism, both chemical sources and sinks exist. A few have no sources or no sinks. This may be intentional (for example, terpenes are destroyed but not produced in atmospheric chemistry mechanisms) but it may also point to missing reactions in the mechanism under construction. With MEXPLORER, it is easy to scan the whole mechanism for such species (called "leaves" in graph theory) with "l_show_primary=True" and "l_show_final=True".

– Graph theory provides many algorithms that can assign key values, either to the whole graph or to individual vertices. These tools can be applied to chemical mechanisms, when they are available as graphs. For example, Silva et al. (2020) used the graph properties modularity and reciprocity to compare mechanisms of different complexity. MEXPLORER provides access to all algorithms which are implemented in graph-tool.

## 2.3 Mechanism evaluation

If turnover rates of individual reactions are available, they can be imported into MEXPLORER and used to evaluate the results of a model simulation.

### 2.3.1 The max-flow problem (finding important reaction pathways)

Often, a key question is not only the magnitude of individual rates but also the overall rate going from a source to a target species. In terms of graph theory, this corresponds to a maximum-flow problem. MEXPLORER allows to choose between three algorithms which are provided by graph-tool:

– Edmonds–Karp algorithm (Edmonds and Karp, 1972)

– Push-relabel algorithm (Goldberg and Tarjan, 1986)

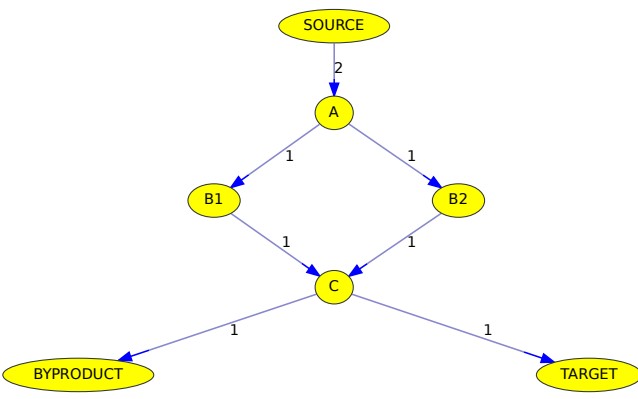

**Figure 6.** Simplified example of the max-flow problem from the source to a target. Edge labels denote reactions rates in arbitrary units. See text for details.

– Boykov-Kolmogorov algorithm (Boykov and Kolmogorov, 2004)

A review by Goldberg and Tarjan (2014) and the graph-tool documentation (https://graph-tool.skewed.de/static/doc/flow.html) provide additional information about these algorithms, including their time complexity (big O notation).

As an example, Fig. 2 shows the reactions leading from $C_5H_8$ (isoprene) to CO. Using the Edmonds–Karp algorithm, the thickness of the arrows indicates the importance of individual pathways.

Another example is shown in Fig. 4 where the reaction sequence from HCHO to $CO_2$ in JAMOC is analyzed. This mechanism contains gas-phase as well as aqueous-phase reactions. The Edmonds–Karp algorithm yields important pathways in both phases.

Although all algorithms determine the maximum flow, the calculated pathways are not necessarily identical. Figure 6 illustrates why the path is not unambiguous when parallel pathways exist: Although the source reacts at a rate of 2 (in arbitrary units), the overall flow from the source to the target is only 1, due to the bottleneck (called minimum cut in graph theory) from C to the target. The maximum flow can either be achieved by going from A to C via B1, or via B2. Thus it is impossible here to judge if B1 or B2 is the more important intermediate.

### 2.3.2 Strongly connected components (chemical "families")

Mechanisms often contain very fast reactions which form catalytic cycles. They may also contain null cycles which neither produce or remove any species, i.e., have a net effect of zero. In atmospheric chemistry, species are often grouped into so-called families (Crutzen and Schmailzl, 1983). They are defined such that reactions inside the null cycles transform species only within the family but don't change the sum

of the family members. Taking the Chapman mechanism as an example again, an "odd oxygen" family can be defined as the sum of O and $O_3$. Neither reaction (R2) nor (R4), which together form a null cycle, change the amount of odd oxygen.

Trying to find important reactions in a complex mechanism can be hampered by fast reactions inside null cycles. Therefore, it is important to detect those, and define suitable families. MEXPLORER can be used to locate fast cycles above a specified threshold in a complex mechanism. Using the command "family=1E-14", MEXPLORER will list all families, in which the members can be interconverted at a rate of $\geq 10^{-14}$. The unit of this value is adopted from the input file with the reaction rates, here it is $mol\,mol^{-1}\,s^{-1}$. When plotting the graph of the mechanism, all reactions inside the families are shown as red arrows. In terms of graph theory, MEXPLORER first creates a subgraph where the edge property "reaction rate" is above the specified threshold. Next, it finds all strongly connected components in the subgraph.

## 3  Summary and outlook

The MEXPLORER software was presented which can be used to analyze, reduce, and visualize complex chemical reaction mechanisms.

Currently, MEXPLORER can only import reaction rates at one selected time step. A planned extension is to read reaction rates at several time steps, e.g., for a whole day. When representing the diurnal variation in reaction rates as a variation in the width of the reaction arrows, it will be possible to produce movies with changing arrow widths, illustrating how important pathways may change throughout the diurnal cycle.

*Code availability.* MEXPLORER is available as a community tool published under the GNU General Public License[5]. The code and the user manual are available at https://doi.org/10.5281/zenodo.8132472. MEXPLORER is also included in version 4.6.0 of the CAABA/MECCA atmospheric chemistry box model at https://gitlab.com/RolfSander/caaba-mecca/-/tree/develop.

*Competing interests.* The author is a member of the editorial board of GMD. The peer-review process was guided by an independent editor, and the author has no other competing interests to declare.

*Acknowledgements.* The code uses the open source software graph-tool (Peixoto, 2014), Python and Graphviz.

---

[5]https://www.gnu.org/licenses/gpl-3.0.html

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
