# Peer review of "MEXPLORER 1.0.0 – A mechanism explorer for analysis and visualization of chemical reaction pathways based on graph theory"

_EGUsphere, 2023_

## Author Response (AR1)

```
* * *
RC1:
* * *
```

> The manuscript by R. Sander presents an open-source code for the
> management of complex reaction mechanisms, employing a graph-based
> representation to enable the utilization of several tools from Graph
> Theory. The capabilities of the software are clearly highlighted,
> mentioning the different functionalities that MEXPLORER provides to
> visualize, filter, build and evaluate complex reaction mechanisms.
> Nonetheless, some of these aspects are not too contextualized, and I
> think that the manuscript could be more complete by providing
> additional details and references beyond this direct showcase of
> functionalities: in their absence, the article seems, in some parts,
> more like a user manual for the program. Consequently, I would suggest
> some major revisions before recommending publication.

I'd like to thank the referee for reviewing the manuscript. My
individual replies are embedded below:

> Line 25. Other possible domains of application of MEXPLORER are
> proposed ("e.g., chemical engineering and marine chemistry"), but no
> references of specific complex reaction mechanisms in these fields are
> given.

I now took a closer look at these other research areas (of which my
current knowledge is limited).

With respect to marine chemistry, I've come to the conclusion that here,
the models are not as complex as in other areas (e.g., Follows et al.,
https://doi.org/10.1016/j.ocemod.2005.05.004). I have removed marine
chemistry from my list of examples. A more general description of
aqueous phase oxidation processes can be found in the paper by Li &
Crittenden (https://doi.org/10.1021/es802039y).

Within chemical engineering, combustion chemistry is arguably the
research field that has to cope with the largest reaction mechanisms,
see e.g., Westbrook et al.
(https://doi.org/10.1016/J.COMBUSTFLAME.2008.07.014). The introduction
section in the book "Analysis of Kinetic Reaction Mechanisms" by Turanyi
& Tomlin (https://doi.org/10.1007/978-3-662-44562-4) provides a good
overview of fields that need complex reaction mechanisms. They mention
combustion, atmospheric chemistry, environmental modelling, process
engineering, and systems biology. Westbrook et al. and Turanyi & Tomlin
are now cited in the manuscript.

> Additionally, I believe that computational chemistry should also be
> mentioned, since many recent efforts are tackling, precisely, the
> automated generation of very complex reaction mechanisms from DFT
> calculations and, consequently, its visualization and treatment.
> Relevant and recent examples could be the work from Prof. Maeda (GRRM:
> [S. Maeda, Y. Harabuchi, H. Hayashi, T. Mita, Annu. Rev. Phys. Chem.,
> 2023, 74], [Y. Sumiya, Y. Harabuchi, Y. Nagata, S. Maeda, JACS Au,
> 2022, 2, 1181-1188) or Prof. Martínez-Núñez (AutoMeKin: [E.
> Martínez-Nuñez, G.L. Barnes, D.R. Glowacki, S. Kopec, D. Pelaez, A.
> Rodriguez, R. Rodriguez-Fernandez, R.J. Shannon, J.J.P. Stewart, P.G.
> Tahoces, S.A. Vazquez, J. Comput. Chem., 2021, 42, 2036], [D.
> Garay-Ruiz, M. Álvarez-Moreno, C. Bo, E. Martínez-Núñez, ACS Phys.
> Chem. Au, 2022, 2, 3, 225-336]), among others.

Yes, it's a good idea to mention this field of research. I have updated the text and added the references suggested by the reviewer.

> Lines 27 – 31. The choice of representing reactions as edges between
> reagents and products is justified as "this produces plots with much
> less visual clutter". While this is correct, to me it seems a bit of
> an oversimplification, reducing graphs to a mere visualization tool. A
> more "conceptual" approach would enrich the discussion, introducing,
> for example, how this reduction in the size of the graph (against
> bipartite graphs as proposed by Silva et al.) may also imply faster
> processing, simplify path finding algorithms, and so on.

MEXPLORER has two main goals: visualization and analysis. With respect to visualization, the reviewer agrees that the current choice (a kinetics graph where reactions are represented by sets of edges from reagents to products) is suitable. With respect to analysis, I would like to note that both types contain the same information as long as the kinetics graph stores the reaction numbers as edge properties. Thus, the same analyses can be performed. The manuscript now describes the interconversion between these types. Speed is not that important for MEXPLORER. Even when using the MCM chemistry, a typical MEXPLORER run finishes within a couple of seconds.

> Line 65. The OIC (overall interaction coefficient) is introduced, but
> it is not explained. The corresponding equation and/or a brief summary
> of how this value can be obtained should be included.

A brief description has been added to the manuscript. In addition, a paper by Niemeyer and Sung is cited now, where the interested reader can find more details.

> 3. Line 71. The affirmation "Alternative representations (...) can be
> created with additional Python functions, if needed" does not seem
> necessary, in the case that it is referred to the users being able to
> write custom functions and expand the code. If it refers instead to
> specific existing functions in MEXPLORER that have not been discussed
> along the section, a brief enumeration of these would be of interest.

> Line 86. Same as point 3.

I see that this needs to be explained better. The sentence refers indeed to custom functions that can be written by the users. I mention this specifically because the modular structure of the code provides entry points to facilitate the implementation of such user-defined Python functions. In contrast, all the options listed above in Section 2.1.1 can be controlled via Python config (*.ini) files, without the need for any code changes. For users with only limited knowledge about the Python language, editing input files should be much easier than generating new code. The manuscript has been adjusted to explain this better.

> Lines 80 – 85. Regarding the function to merge parallel edges, another
> possible simplification would be to take the "best" reaction of the
> bunch as the representative of the group (e.g. the one with the
> largest rate constant), so a simpler graph can be obtained where every
> edge is conceptually a single reaction, with a single set of names and
> parameters. Is this available in some way/is there a reason not to do
> it?

This is an interesting idea, and it would be worthwhile to implement it. However, it would be necessary to consider a couple of points first:

- Selecting the "best" reaction is not unambiguous. Different results will be obtained when comparing nighttime and daytime chemistry, or when looking at different scenarios (e.g., urban vs rural).

- This method can only be applied when reaction rates are available, e.g., when analyzing the output of a model simulation.

The suggested method would be a good addition for specific use cases. I think it is a good example where the user could write their own code for their task.

> Line 102. Having a filtering function for the specific JAMOC mechanism
> seems a bit specific. Adding some more context on the importance of
> this could make it clearer.

I agree that this is a very specific case. To provide more background, a reference to Rosanka et al. (2021), which describes the JAMOC mechanism, has been added here. In addition, it is now mentioned that the JAMOC filter can also serve as a template when a new, user-defined function needs to be written for a specific task.

> Line 139 (max-flow problem section). While the available algorithms
> are properly listed and referenced, given the importance and
> difficulty of the general problem of locating important reaction
> pathways in any complex mechanism, a more thorough discussion would be
> quite interesting. For instance, introducing the key differences
> between the three available algorithms, as well as an idea on the
> timings that they take for increasingly complex networks. Enriching
> this part of the discussion would reinforce the manuscript as a whole.

I agree: Locating important pathways in complex mechanisms is both important as well as difficult, and it deserves more discussion. However, the max-flow algorithms are part of graph-tool, not MEXPLORER, and the mathematical theory behind the algorithms has already been discussed extensively in the literature. I therefore think it is more appropriate to refer to the available literature for additional information. A review article (Goldberg and Tarjan, 2014) and the graph-tool documentation at https://graph-tool.skewed.de/static/doc/flow.html are now cited. In the text of the manuscript, I'd prefer to focus on the application to reaction mechanisms. In particular, it is now mentioned that these algorithms can find the maximum flow, but they cannot provide a unique answer about the pathway. This is illustrated with an example.

> Lines 167 – 169. The wording "movies with changing arrow widths,
> visualizing how important pathways may change through the diurnal
> cycle" hints more at a very specific example than to a general feature
> of the code such as the time evolution of reaction networks and the
> corresponding representation. I would suggest rephrasing the paragraph
> to first comment on the idea of network evolution, then on the
> possibility of plotting movies, and only then to the specific
> day-cycle idea.

The outlook section mentions "movies with changing arrow widths" specifically, because this task is high in the priority list of new features for future versions of MEXPLORER. It is currently not planned to consider evolving networks.

> Figure sizes are quite inconsistent. While it is understandable that

> this kind of automatically generated complex graphs might have varying
> shapes and sizes, a general consistency would be desirable. For
> instance, Figures 4 and 5 could be reduced so the general node and
> font sizes match Figures 2 and 3.

I have to admit that I have so far not made any efforts to produce
uniform node and font sizes for all figures. The reason is that final
publications in GMD are typeset in a two-column format in which some
figures occupy both columns, and some just one. In the latter case, the
size of the figure changes considerably compared to the layout of the
manuscript. I will try to produce equal node and font sizes as much as
possible, once the paper is typeset in its final layout.

> Some of the node colors are a bit dark to allow the text to be
> comfortably read, especially in smaller sizes. Figure 3 is
> particularly hard to read, as it is composed by green and turquoise
> nodes that encounter this problem. Using paler shades of these two
> colors will make the final result much more readable.

As suggested, the colors have been changed for better readability.
* * *
RC2:
* * *
> R. Sander presents MEXPLORER, a Python-based tool using graph
> theoretical algorithms for chemical mechanism visualization and
> analysis. The manuscript is concise and well written: I have only
> fairly minor comments on the manuscript itself. The idea and potential
> value of the software is also clear. However, some major issues in the
> software need to be addressed before I can recommend its publication —
> these points are detailed in specific comments on the software. If
> MEXPLORER is being published as an open-source tool, then changes
> should be made to improve its accessibility. Right now, barriers to
> installation and some incompatibilities with standard KPP formats that
> might prevent it from becoming a community tool. The idea of
> KPP-compatible mechanism visualization and analysis is potentially
> quite valuable to the atmospheric chemistry community and I hope the
> recommended changes help facilitate adoption of MEXPLORER as a useful
> tool.

I'd like to thank the referee for the review. In particular, I would
like to thank them for not only reviewing the manuscript but also
testing the MEXPLORER code!

Based on the suggestions, I have created the new version which solves
the issues raised by the reviewer. Unfortunately, however, two
additional problems were found while testing MEXPLORER with KPP files
from other models (negative products and multiline equations, see
below). These could not be solved but they are now mentioned in the user
manual, and simple workarounds are offered.

My individual replies are embedded below:

> Community model — would it perhaps be more accurate to call this a
> community package or tool? Model is fine here if the author thinks
> that is most appropriate

I agree that "tool" is better than "model", and I have changed the text
accordingly.

> Reactions as edges seems inaccurate or at least incomplete. Table 1
> and Figure 1 show that multiple edges correspond to the same reaction
> — it may be more accurate to describe reactions with this graph
> structure as "sets of edges"

The reviewer is correct. I have changed the text accordingly.

> Lines 27-30, I agree with Reviewer 1's point 2: more motivation for
> the choice of this specific graph architecture here might be
> appropriate, and discussion of the differences. Is another advantage
> of this graph architecture faster algorithms than the bipartite graph?
> Is this type of graph more common than a bipartite graph? Are there
> any disadvantages or information lost?

I'm not sure which type is currently more common in the literature. It
seems that the "kinetics graph" as used by MEXPLORER has been used for a
long time whereas the bipartite "species-reaction graph" appears mostly
in more recent publications. Please see also my answer to the related
question by reviewer #1.

> Lines 27-30 It seems that the reaction splitting might come from this
> choice of graph architecture — here, I would also rephrase to
> "…reactions are represented as sets of edges pointing…" and do this
> throughout the manuscript

This is done now throughout the manuscript.

> Line 123 I believe the words "for example" can be removed here

Done.

> Line 161 1e-14 — what units are these interconversions in?

The unit of this value is adopted from the input file with the reaction
rates, here it is mol/(mol*s). This is now also mentioned in the
manuscript.

> Step by step installation instructions may be important. See the
> following specific comments on the software. 1. MEXPLORER did not work
> out of the box. It would be good to explicitly list all dependencies
> in the manual.

I agree. A list of dependencies is now provided in the file
requirements.txt. This is mentioned in the updated manual.

> I had not used graph-tool before, and was unable to install it into my
> standard conda environment (pip is not listed as an option on the
> graph-tool installation instructions:
> https://git.skewed.de/count0/graph-tool/-/wikis/installation-instructions).

They reason why graph-tool is very fast is because it is based on a C++
library (wrapped in Python). The downside of this architecture is that
it cannot be installed with a simple pip command because of the C++
dependencies (https://graph-tool.skewed.de/performance).

> The graph-tool documentation states that attempting to install into an
> existing environment might just hang/freeze, so I created a new
> environment for this project:
> conda create --name mexplorer -c conda-forge graph-tool

> conda activate mexplorer
> I am using python 3.12.0 and installed graph-tool version 2.58.
> 2. After setting up a new environment for graph-tool to run MEXPLORER,
> attempting to install the netCDF4 library with conda introduces
> conflicts and installation is cancelled. A workaround with pip was
> possible. If the installation process was very easy, then such an
> abbreviated installation section in the MEXPLORER user manual would
> make sense; however, the details of graph-tool installation and
> incompatibilities of several of the packages mean that more detailed
> installation instructions should be given, perhaps as KPP does:
> https://kpp.readthedocs.io/en/stable/getting_started/01_installation.html

I'm not the author of graph-tool, therefore my ability to solve
installation problems is quite limited. I agree, however, that I should
provide more information where to find help. In the user manual, I am
now listing several options. This includes the
"installation-instructions" web page that the reviewer has already
mentioned, and also the discourse forum and the issue tracker. In
addition, an issue can be opened on my CAABA/MECCA gitlab page for
MEXPLORER-related questions at
https://gitlab.com/RolfSander/caaba-mecca/-/issues.

> 3. Once installing all the required dependencies, I attempted to run
> the first example from the manual as ./mexplorer.py chapman.ini, but
> got the following message while plotting in
> graph_tool/draw/graphviz_draw.py
> line 316, in graphviz_draw
>     if gv_new_api:
>        ^^^^^^^^^^
> NameError: name 'gv_new_api' is not defined
> graphviz_draw.py, line 589, in graphviz_draw
>     libgv.agclose(gvg)
>     ^^^^^
> NameError: name 'libgv' is not defined. Did you mean: 'libc'?
> It seems like a related issue is on the git documentation:
> https://git.skewed.de/count0/graph-tool/-/issues/767, though the
> proposed workaround is already implemented on line 8 of
> define_graph.py.

Without additional information about the operating system and the
available software on the computer of the reviewer, it is difficult for
me to analyse this problem. My guess is that graphviz is not installed
properly. On a linux system, I would try:

sudo apt install graphviz

If I understand the graph-tool package correctly, it depends on the
graphviz software, and an installation of graph-tool will automatically
install graphviz as well. I do not understand how the reviewer was able
to install graph-tool without graphviz.

> 4. ./mexplorer.py CH4_def.ini worked out of the box.

Using CH4_def.ini as input, MEXPLORER takes the input file for CH4 and
defines the corresponding graph, which is then saved in CH4.xml.gz. This
part of MEXPLORER does not depend on graphviz, which may explain why
CH4_def.ini works for the reviewer but the other examples don't.

> 5. All 3 other plotting examples did not work out of the box, with a
> similar issue to issue 3: mom_C5H8_to_CO_maxflow.ini, mcm_S.ini,

> mom_pyruvic_acid.ini

I'd be happy to help but, as mentioned above, it is difficult to analyse
this problem without additional information. To preserve the anonymity
of the reviewer, maybe the editor could relay a more detailed problem
description to me?

> 6. Just a small suggestion: Lines 45 and 65 on define_graph.py assume
> the spc files and eqn files are named "mecca" — is this necessary?
> Could these lines just find the .eqn and .spc files, which might be
> named after the mechanism (e.g. SAPRC99) rather than named to mecca?

Initially, I had assumed that it would be sufficient if the users simply
rename their input files to mecca.spc and mecca.eqn. However, to provide
a more user-friendly solution, I have now added the option to specify
these names in the config file. For example, the new config file
boxmox_def.ini contains the following two lines to load the smog
mechanism of BOXMOX:

spcfilename = smog.spc
eqnfilename = smog.eqn

> 7. To my knowledge, .spc files aren't required for KPP to work — KPP
> can parse species from .eqn files. Some communities that do use KPP
> don't use spc files, such as GEOS-Chem users:
> https://github.com/geoschem/geos-chem/tree/main/KPP/fullchem. It seems
> like spc files, though nice for mass balance checking, might prevent
> broader adoption. Would it be better to make .spc files an option,
> rather than a requirement?

KPP users can choose between two options: Either the species and the
equations are in separate files (called *.spc and *.eqn), or they are
within one merged file. In the latter case, the combined file can be
used as input for both in MEXPLORER. For example, the definitions of
species and equations in the config file for GEOS-Chem can point to the
same file:

spcfilename = fullchem.eqn
eqnfilename = fullchem.eqn

While testing MEXPLORER with input from GEOS-Chem, I encountered two
additional problems: negative products and multiline equations. Although
rare, they are syntactically legal in KPP. As they cannot be parsed by
MEXPLORER, I have now explained these limitations at the end of section
3.1 in the user manual, and a workaround is offered.

> 8. Other mechanisms that have .eqn and .spc files do not work with
> MEXPLORER. MEXPLORER does not recognize species from several
> mechanisms in the BOXMOX example library (BOX MOdel eXtension to KPP;
> Knote et al., 2015):
> https://mbees.med.uni-augsburg.de/gitlab/mbees/boxmox/-/tree/master/models?ref
> _type=heads.
> Using the NO_NO2 .eqn and .spc files results in empty vertices and
> edges in the .xml.gz output; this issue was reproduced for several
> other larger mechanisms as well.

I'd like to thank the reviewer for pointing out these problems. I have
now tried the BOXMOX files myself, and I discovered two problems: First,
the BOXMOX files contain a mix of tabs and spaces, whereas MEXPLORER
expected only spaces. Second, I had falsely assumed that there would

always be an equation tag (e.g., "<R1>") at the start of each reaction. This is not the case, however, for the reactions in NO_NO2.eqn and other BOXMOX equation files. Both problems are fixed in the latest version of MEXPLORER.

> Much of the difficulty in MEXPLORER does not seem to originate from
> the software itself, but instead in getting a working environment with
> a version of graph-tool that can plot with graphviz (points 1-5). For
> this reason I would suggest either a) improving the documentation for
> installing dependencies or b) ensuring compatibility by shipping all
> dependencies with it, perhaps using a Docker approach.

a) The documentation has been improved as suggested.

b) I've never used docker myself but from what I know about it, this
   could be a good solution. It is certainly something to keep in mind
   for the release of the next version of MEXPLORER.

   Also, I noticed that graph-tool already offers docker as an
   installation option:

   https://git.skewed.de/count0/graph-tool/-/wikis/installation-instructions#installing-using-docker

   Would it be possible to install graph-tool via docker and then simply
   copy the MEXPLORER files into the docker container?
* * *

---

## Author Response (AR2)

> * (from Reviewer #2): "Page 1, line 20: automatized -- perhaps
>   automated?"

Changed as suggested.

> * (from Reviewer #1): "Possibly my wording for 'evolving networks' was
>   not the most adequate. I was referring, more than to the network
>   itself changing, to the changes in the reaction rates mentioned in
>   the manuscript. Thus, I still find the sentence to not convey well
>   the actual chemical meaning of the proposed representation. I'd
>   propose something on the lines of 'this variation in reaction rates
>   can be represented as a variation in the width of reaction arrows,
>   which could be used to produce movies illustrating how important
>   pathways may change throughout the diurnal cycle', reminding the
>   reader of how arrow widths relate to the situation that is modeled."

I think I now understand the suggestion of the referee, and I have
changed the text in the outlook section accordingly.

> * In the reply to the reviews, a new version of the code addressing
>   points raised by the reviewers is mentioned. However, the Zenodo
>   link included in the manuscript still leads to the July-2023
>   v1.0.0-rc.0 release - please correct.

During the revision process, I have developed the code only on gitlab,
but I didn't create any new versions on Zenodo. Now, as we are reaching
the end of the review procedure, I have uploaded the revised code to
Zenodo.

>   Also, the statement in the "Code availability" section that
>   "MEXPLORER will also be included in the next version of the
>   CAABA/MECCA" should be changed to point to a specific archived
>   version (or to clearly indicate that it is just a plan, though I
>   would discourage publishing such statements).

It is mentioned now that MEXPLORER is also available in CAABA/MECCA
version 4.6.0.

>   The linked repository at https://gitlab.com/RolfSander/caaba-mecca
>   does not seem to include any of the MEXPLORER files archived on
>   Zenodo.

I forgot to mention that the files are available in the "develop"
branch. Maybe you looked at the "master" branch when you tried to find
them?
* * *
In addition, I had to make a few minor changes with respect to the MCM
chemical mechanism. Since January 2024, the MCM has a new web page and a
new export function for files in KPP syntax. A few adjustments were made
to make MEXPLORER compatible with the new MCM-generated KPP files. As a
result, the regenerated Fig. 3 has a new layout (but still contains the
same reactions).